# High-Precision Depth Map Estimation from Missing Viewpoints for 360-Degree Digital Holography

Hakdong Kim [1], Heonyeong Lim [1], Minkyu Jee [2], Yurim Lee [3], MinSung Yoon [4,*]
and Cheongwon Kim [5,*]

1   Department of Digital Contents, Sejong University, Seoul 05006, Korea
2   Department of Software Convergence, Sejong University, Seoul 05006, Korea
3   Department of Artificial Intelligence and Linguistic Engineering, Sejong University, Seoul 05006, Korea
4   Communication & Media Research Laboratory, Electronics and Telecommunications Research Institute, Daejeon 34129, Korea
5   Department of Software, Sejong University, Seoul 05006, Korea
*   Correspondence: msyoon@etri.re.kr (M.Y.); wikim@sejong.ac.kr (C.K.)

**Abstract:** In this paper, we propose a novel model to extract highly precise depth maps from missing viewpoints, especially for generating holographic 3D content. These depth maps are essential elements for phase extraction, which is required for the synthesis of computer-generated holograms (CGHs). The proposed model, called the holographic dense depth, estimates depth maps through feature extraction, combining up-sampling. We designed and prepared a total of 9832 multi-view images with resolutions of 640 × 360. We evaluated our model by comparing the estimated depth maps with their ground truths using various metrics. We further compared the CGH patterns created from estimated depth maps with those from ground truths and reconstructed the holographic 3D image scenes from their CGHs. Both quantitative and qualitative results demonstrate the effectiveness of the proposed method.

**Keywords:** depth map estimation; 360-degree holographic contents; deep learning



## 1. Introduction

A depth map image represents information related to the distance between the camera's viewpoint and the object's surface. It can be reconstructed based on the original (RGB color) image and generally has a grayscale format. Depth maps are used in three-dimensional computer graphics, such as three-dimensional image generation and computer-generated holograms (CGHs). In particular, phase information, which is an essential element for CGHs, can be acquired from depth maps [1,2]. The 360-degree RGB images and their corresponding depth map image pairs are required to observe 360-degree digital holographic content. If a specific location does not have a depth map (missing viewpoint), its corresponding holographic 3D (H3D) scene will not be visible. Depth map estimation compensates for missing viewpoints and contributes to the formation of realistic 360-degree digital hologram content. In this study, we propose a novel method for learning depth information from captured multi-view points and estimating depth information from various missing viewpoints. An overall schematic of the proposed multi-view method is shown in Figure 1.

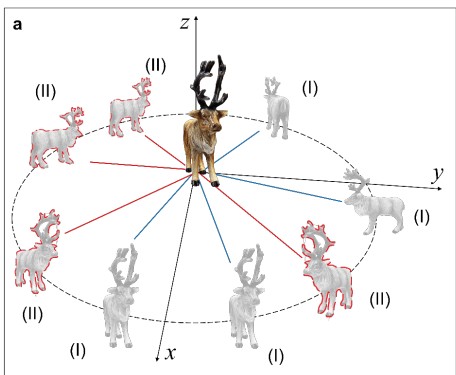 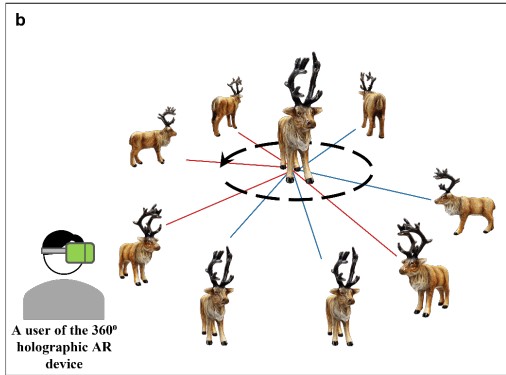

**Figure 1.** (**a**) Depth map estimation at missing viewpoints (II) using multi-viewed RGB-Depth map set (I). Only RGB color image exists at each missing viewpoint. (**b**) The proposed HDD model provides holographic 3D content (CGHs) with even missing viewpoints (II) covered to the full so that the user can watch much richer 360-degree holographic AR movies (3D model of the deer in figure was provided from TurboSquid, https://www.turbosquid.com/ko/3d-models/free-caribou-3d-model/591531, accessed on 17 September 2022).

## 2. Previous Research

Many previous studies have estimated depth maps based on three distinct approaches. The first is the monocular approach. Battiato et al. [3] proposed the generation of depth maps using image classification. In this work, digital images are classified as indoor, outdoor, and outdoor with geometric objects, with low computational costs. Eigen et al. [4] proposed a convolutional neural network (CNN)-based model consisting of two different networks. One estimates the global structure of the scene, whereas the other estimates the local information. Koch et al. [5] studied the preservation of edges and planar regions, depth consistency, and absolute distance accuracy from a single image. Other studies adopting conditional random fields [6–8], generative adversarial networks [9,10], and U-nets with an encoder–decoder structure have been introduced [11]. Alhashim et al. [12] applied transfer learning to a high-resolution depth map estimation, which we refer to as conventional dense depth (CDD) for the remainder of this paper.

The second method is a stereo-view approach, which uses a pair of left and right color images as input [13,14]. Wang et al. [15] recently applied self-supervised learning to single depth map estimation based on stereoscopic color images. Their model used a pair of left and right color images to synthesize a middle-view-pointed color image and estimated its corresponding depth map. The stereo-view approach is different from the proposed multi-view method as it can estimate the depth map of only a single-viewed image located between the left and right images.

The third approach estimates a single depth map using multi-view images (more than two RGB images). Many of them use the plane-sweep method [16], which is a basic geometry algorithm for finding intersecting line segments. Choi et al. [17] used CNNs for multi-view stereo matching, which combines the cost volumes with the depth hypothesis in multi-view images. Im et al. [18] proposed an end-to-end model that learns a full plane-sweep process, including the construction of cost volumes. Recently, Zhao et al. [19] proposed an asymmetric encoder–decoder model that has improved accuracy for outdoor environments. Wang et al.'s work features a CNN for solving the depth estimation problem on several image–pose pairs that are taken continuously while the camera is moving [20]. All of the above studies can generate only a single depth image using multi-view RGB images.

To provide realistic holographic metaverse [21,22], Augmented Reality (AR) [23], or Virtual Reality (VR) [24] content (see Figure 1b) to users, it is essential to estimate as many highly precise depth maps as possible in a short time for each given narrow angular range. Therefore, in this study, although we use the input data of multi-view RGB images, we do not adopt either conventional stereo-view or multi-view methods because these methods

use multiple RGB images as input to output only a single depth map. Our model can generate new depth maps derived from new RGB images of the missing viewpoints, as illustrated in (II) of Figure 1a. As a result, we were able to obtain 360-degree depth maps of objects for realistic holographic 3D or AR/VR content. Furthermore, we performed experiments to test the quality of the estimated depth maps by reconstructing holographic 3D scenes from the CGHs generated by the estimated depth maps. Our experimental results show the effectiveness of our approach.

## 3. Proposed Method

### 3.1. Data Preparations

We first introduce a method that generates a 360-degree, multi-view RGB image–depth map pair data set using the Z-depth rendering function provided by 3D graphic software, Autodesk Maya 2018 [25]. Second, we present a neural network architecture that estimates depth maps. Finally, the experimental results are discussed. In addition, we also show the results of synthesizing CGHs, numerical reconstruction, and optical reconstruction using RGB image–depth map pairs in the next section. In this study, a data set of multi-view RGB image–depth map pairs was generated using Z-depth rendering provided by Maya software. To extract the RGB image–depth map pairs in Maya software, we devised two identical 3D objects located near the origin and a virtual camera with a light source that shoots these two solid objects, so that the virtual camera could acquire depth difference information between two solid figures during camera rotation around the origin. Depth measurements along the Z-direction from the camera were made using a luminance depth preset supplied by Maya software.

The virtual camera acquires 1024 pairs (for each shape of both RGB images and depth maps) while rotating 360-degrees around the axis of rotation. Shapes of 3D objects that we used for the study were a torus, cube, cone, sphere, dodecahedron, and icosahedron. We used a total of 9832 images for the experiments in this study. Among them, 4916 are RGB images and the remaining 4916 are depth map images. Both RGB images and depth map images are classified into six shapes (torus, cube, cone, sphere, dodecahedron, and icosahedron). The first four shapes (torus, cube, cone, sphere) are contained in data set 1 and were used in the first experiment where the depth map estimations were used during the training phase. Of the total data, 60% were used for training and 40% were used for the tests where each instance consists 1024 views. Four objects in data set 1 were learned together without being separated.

The remaining shapes (dodecahedron and icosahedron) were used in the second experiment where the estimations for their objects were only used for testing purposes. We conducted this experiment to demonstrate our proposed method's generalizability in unseen shapes. Table 1 shows the camera settings used to create 3D objects, and Table 2 shows the data specification for the entire experiment.

**Table 1.** Camera settings for 360-degree 3D content.

| | |
|---|---|
| Distance from virtual camera to 255 depth | 11 cm |
| Margin from depth boundary to object | 2 cm |
| Distance from virtual camera to 0 depth | 28.7 cm |
| Distance between two objects (Center to center) | 8.3 cm |
| Distance from 0 depth to 255 depth | 14.2 cm |
| Radius of camera rotation path (R) | 20 cm |

**Table 2.** Data sets in which RGB images and depth maps are prepared for training and test in the research.

| Data Set 1 (Depth Map Estimation for Objects Used in Train) | | | | | | |
|---|---|---|---|---|---|---|
| Generated sample | | Shape | | Training set/Test set | | |
| RGB images | 4096 | Torus | 1024 | Set for training | 614 | Set for test | 410 |
| | | Cube | 1024 | | 614 | | 410 |
| | | Cone | 1024 | | 614 | | 410 |
| | | Sphere | 1024 | | 614 | | 410 |
| Depth map images | 4096 | Torus | 1024 | Set for training | 614 | Set for test | 410 |
| | | Cube | 1024 | | 614 | | 410 |
| | | Cone | 1024 | | 614 | | 410 |
| | | Sphere | 1024 | | 614 | | 410 |
| Data set 2 (Depth map estimation for objects not used in train) | | | | | | |
| Generated sample | | Shape | | Training set/Test set | | |
| RGB images | 820 | Dodecahedron | 410 | Set for training | - | Set for test | 410 |
| | | Icosahedron | 410 | | - | | 410 |
| Depth map images | 820 | Dodecahedron | 410 | Set for training | - | Set for test | 410 |
| | | Icosahedron | 410 | | - | | 410 |

### 3.2. Proposed Model

The proposed depth map estimation model HDD (Holographic Dense Depth) consists of two components: the encoder and decoder, as shown in Figure 2 and Table 3. We experimented with several encoders as seen in previous studies [4,8,12]. Then, we found that deeper layers and more convolution operations do not necessarily lead to better depth map estimation results and DenseNet-161 [26] is an optimal model for our purpose. The encoder performs feature extraction and down-sampling of the input RGB images. Each feature maps are connected by skip-connections to correspond the up-sampling layer in the decoder. At this point, both layers are calculated as concatenation. The decoder performs up-sampling by concatenating the extracted features based on the size of the RGB image. The weights for both components are optimized by a loss function that minimizes the discrepancy between the ground truth and the estimated depth map. CDD learns and estimates the depth from a single viewpoint. On the other hand, the proposed model, HDD, learns depth from multiple viewpoints and estimates the depth of viewpoints that are not used for training (new viewpoints). We also adopted bilinear interpolation in the up-sampling layer and ReLU as the activation function, of which are different from CDD. Consequently, we obtained better depth estimation results.

Since Eigen et al.'s research [4]—which was an early CNN-based depth map estimation study—in general, the loss function of depth map estimation has been based on a difference between ground truth depth map and estimated depth map. Furthermore, in the comparative model CDD [12], *MSE* (Mean Squared Error) and *SSIM* (Structural Similarity) [27] were used together for loss function. They optimized coefficient of loss function for CDD's purpose (general depth map estimation). In this study, we experimentally explored the best coefficient of loss function that gives the optimal depth estimation for our purpose (depth map estimation for 360-degree holographic content). As a result from this experiment, we concluded that the coefficient of 100% *MSE* and 0% *SSIM* is optimal for depth map estimation for our data, of which are suitable for 360-degree holographic content. The details of the loss function coefficient are shown in Figure S1.

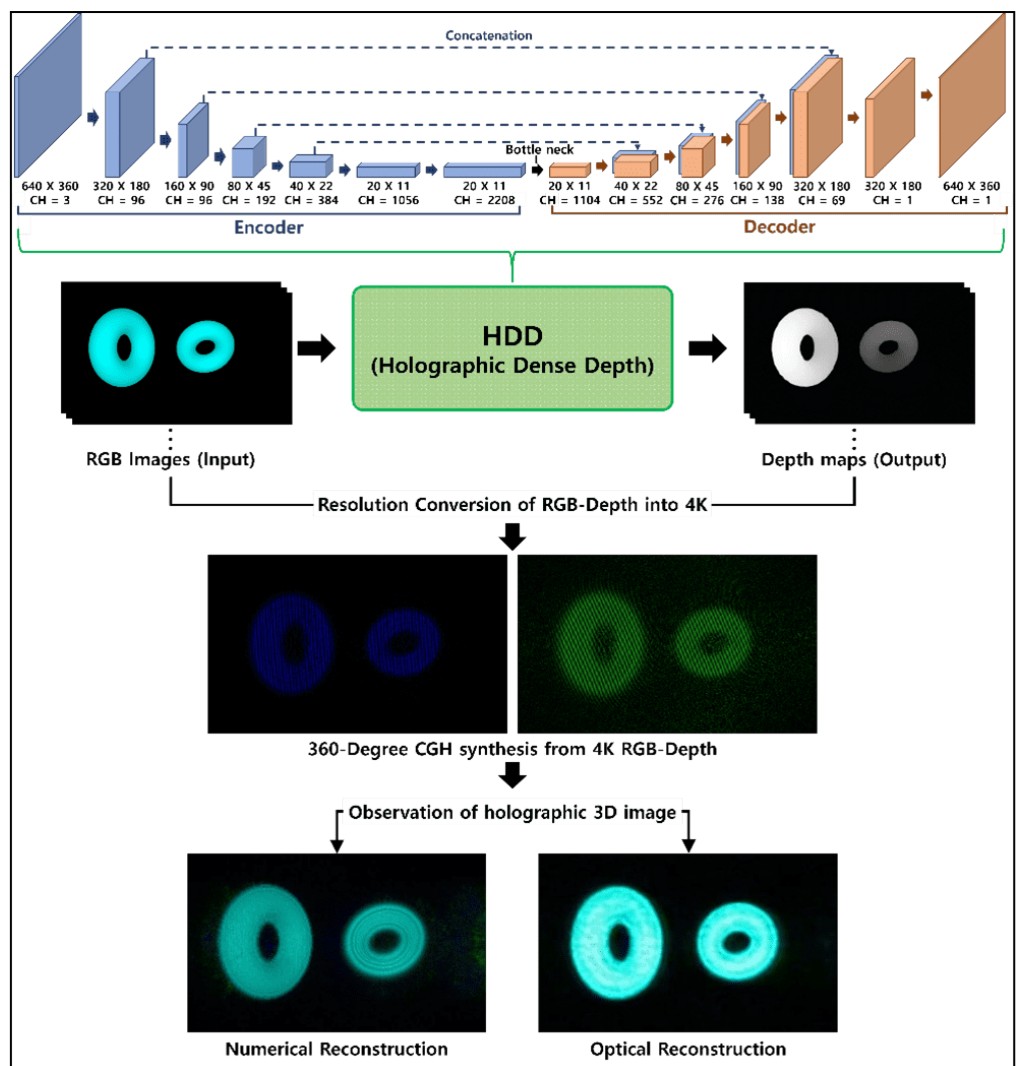

**Figure 2.** Pipeline for proposed model HDD and digital hologram reconstruction. Data set for RGB color image and ground truth depth map is created using Autodesk Maya 2018, Maya Software (https://www.autodesk.com/products/maya/overview, accessed on 17 September 2022).

**Table 3.** Main components in the proposed model HDD.

| Architecture of Proposed Model | | |
|---|---|---|
| Input | Input (RGB images) | [ch = 3, shape = 640 × 360] |
| Convolution | 7 × 7 convolution, stride 2 | [ch = 96, shape = 320 × 180] |
| Encoder (Pre-trained DenseNet-161) | Batch normalization<br>ReLu<br>3 × 3 max pooling | [ch = 96, shape = 160 × 90] |
| | Dense block (6 dense layers)<br>transition layer | [ch = 192, shape = 80 × 45] |
| | Dense block (12 dense layers)<br>transition layer | [ch = 384, shape = 40 × 22] |
| | Dense block (36 dense layers)<br>transition layer | [ch = 1056, shape = 20 × 11] |
| | Dense block (24 dense layers) | [ch = 2208, shape = 20 × 11] |
| | Batch normalization | [ch = 2208, shape = 20 × 11] |

**Table 3.** *Cont.*

| Bottleneck | 1 × 1 convolution | [ch = 1104, shape = 20 × 11] |
|---|---|---|
| Decoder (Dense Depth) | Up-sampling layer | [ch = 552, shape = 40 × 22] |
| | Up-sampling layer | [ch = 276, shape = 80 × 45] |
| | Up-sampling layer | [ch = 138, shape = 160 × 90] |
| | Up-sampling layer | [ch = 69, shape = 320 × 180] |
| Convolution | 3 × 3 convolution | [ch = 1, shape = 320 × 180] |
| Output | Bilinear interpolation | [ch = 1, shape = 640 × 360] |
| **Dense layer** | **Transition layer** | **Up sampling layer** |
| Batch normalization | Batch normalization | Bilinear interpolation |
| ReLu | ReLu | Skip connection |
| 1 × 1 convolution | 1 × 1 convolution | 3 × 3 convolution |
| Batch normalization | 2 × 2 max pooling | Batch normalization |
| ReLu | - | ReLu |
| 3 × 3 convolution | - | 3 × 3 convolution |
| - | - | Batch normalization |
| - | - | ReLu |

## 4. Experiment Results and Discussion

All experiments were performed in the following hardware environment: ASUS ESC8000-G4 series with Nvidia Titan RTX × 8.

### 4.1. Quantitative Results

Figure 3a shows the loss curve averaged over four objects (torus, cube, cone, and sphere) during training for the HDD model. Then, Figure 3b shows the typical trends of *PSNR* (peak signal-to-noise) of HDD and CDD for the torus, where the *x*-axis represents the step number over 90 epochs. In the case of the torus, Figure 3c shows the distribution of *ACC* (accuracy) [28], indicating the similarity between the ground truth depth map and the depth map estimated from HDD and from CDD, where the *x*-axis represents the angular degree corresponding to viewpoints (see Figure 1). In addition, Figure 3d shows the comparison between *RMSE* from the HDD and *RMSE* from CDD for each object. Table 4 shows the performance comparison between HDD and CDD by using various quantitative metrics, for each one that is defined and explained in Equations (S1)–(S9).

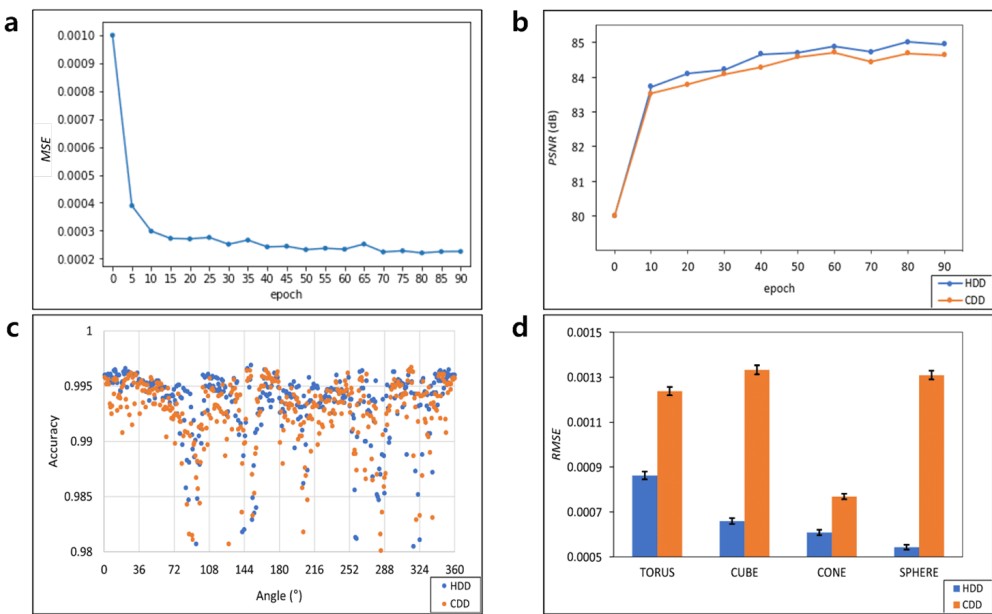

**Figure 3.** (**a**) Curve of the loss *MSE* averaged over all objects during training for the HDD model; (**b**–**d**) comparison of HDD and CDD using *PSNR* trend, *ACC* distribution with ground truth depth map for torus, *RMSE* difference for all objects after training.

**Table 4.** Quantitative comparison of HDD (proposed model) and CDD (reference model). (**a**–**c**): The higher is better; (**d**–**g**): The lower is better.

| Models | (a) *SSIM* | | | | (b) *PSNR* (dB) | | | |
|---|---|---|---|---|---|---|---|---|
| | Torus | Cube | Cone | Sphere | Torus | Cube | Cone | Sphere |
| HDD | 0.9999 | 0.9999 | 0.9999 | 0.9999 | 84.95 | 84.42 | 84.90 | 85.03 |
| CDD | 0.9999 | 0.9999 | 0.9999 | 0.9999 | 84.64 | 83.94 | 84.68 | 84.62 |

| Models | (c) *ACC* | | | | (d) *Abs rel* | | | |
|---|---|---|---|---|---|---|---|---|
| | Torus | Cube | Cone | Sphere | Torus | Cube | Cone | Sphere |
| HDD | 0.9933 ± 0.0040 | 0.9933 ± 0.0030 | 0.9928 ± 0.0037 | 0.9965 ± 0.0012 | 0.022 | 0.018 | 0.022 | 0.017 |
| CDD | 0.9925 ± 0.0039 | 0.9934 ± 0.0027 | 0.9926 ± 0.0036 | 0.9959 ± 0.0013 | 0.019 | 0.017 | 0.016 | 0.017 |

| Models | (e) *Sq rel* | | | | (f) *RMSE* | | | |
|---|---|---|---|---|---|---|---|---|
| | Torus | Cube | Cone | Sphere | Torus | Cube | Cone | Sphere |
| HDD | 0.0058 | 0.0046 | 0.0058 | 0.0043 | 0.0009 | 0.0007 | 0.0006 | 0.0005 |
| CDD | 0.0062 | 0.0052 | 0.0061 | 0.0047 | 0.0012 | 0.0013 | 0.0008 | 0.0013 |

| Models | (g) *LRMSE* | | | |
|---|---|---|---|---|
| | Torus | Cube | Cone | Sphere |
| HDD | 0.0114 | 0.0117 | 0.0111 | 0.0110 |
| CDD | 0.0116 | 0.0122 | 0.0113 | 0.0112 |

### 4.2. Qualitative Results of the Image Quality

#### 4.2.1. Comparison of the Proposed Model (HDD) with the Ground Truth Using Holographic 3D Images Reconstructed from CGH

Figure 4 shows the comparison between 3D images to be constructed from CGH using estimated and ground truth depth maps, respectively. The 3D holographic images were reconstructed by two different methods—numerical and optical reconstruction.

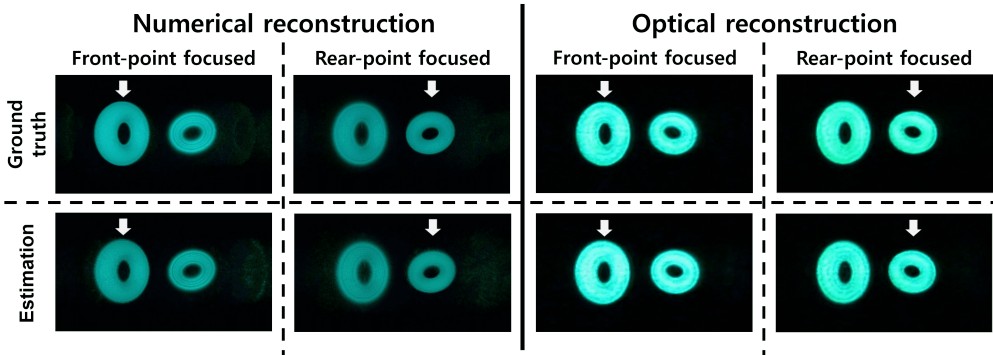

**Figure 4.** Results of numerical and optical reconstruction from CGHs using estimated depth maps as well as ground truth for the torus. A focused object is indicated by an arrow mark. (Results for entire viewpoints are shown in Videos S1 and S2. Results of numerical and optical reconstruction with other objects can be seen in Figure S2).

The CGH image of each viewpoint is calculated by the Fast Fourier Transform (FFT) algorithm [29,30] using RGB images and depth maps. During the FFT computational process, either a pair of an RGB image and its corresponding ground truth or estimated depth map is used as input per viewpoint. The function of CGH *H(x,y)* synthesized from the the FFT-based algorithm contains complex numbers in general and can be written as

$$H(x,y) = |H(x,y)|e^{i\Phi(x,y)} \tag{1}$$

where $|H(x,y)|$ is the amplitude of *H(x,y)* and $\Phi(x,y)$ is its phase, respectively. After the CGHs of 1024 views were prepared for each solid object, an additional process of encoding called Lee's scheme [29] was adapted so that they could be represented on an amplitude-modulating spatial light modulator (SLM), that is, a reflective LCoS (liquid crystal on silicon)-SLM that we used in the study. Lee's method decomposes a complex-value field into four real and non-negative coefficients, $L_m(x,y)$ where *m* is the natural number of $1 \leq m \leq 4$; the decomposition of the hologram function by Lee's encoding can be expressed by

$$H(x,y) = L_1(x,y)e^{i0} + L_2(x,y)e^{i\pi/2} + L_3(x,y)e^{i\pi} + L_4(x,y)e^{i3\pi/2} \tag{2}$$

where at least two among four coefficients $L_m(x,y)$ are equal to zero. The SLM that is used for optical reconstructions can display an 8-bit grayscale with a resolution of $3840 \times 2160$ pixels, an active diagonal length of 0.62", and a pixel pitch of 3.6 μm. A fiber-coupled, combined beam from RGB laser sources (wavelengths: 633 nm, 532 nm, and 488 nm of the MatchBox Laser Series) passed through an expanding/collimating optical device to supply coherent, uniform illumination on the active area of the SLM. A field lens (focal length: *f* = 50 cm) was positioned just after the LCoS-SLM, and an experimental observation of the optically reconstructed images was performed using a DSLR camera (Canon EOD 5D Mark III), whose lens was located within an observation window that was generated near the focus of the field lens [30] (see Figure 5). The results of the camera-captured optical reconstructions and numerical reconstructions from the synthesized CGHs are shown in Figure 4. To prove the depth difference in a real 3D space between two objects based on the prepared 360-degree holographic content, we demonstrated the accommodation effect with

optically realized holographic 3D scenes. Each photograph in the optical reconstruction's column of Figure 4 presents the experimental demonstration to simultaneously indicate the clear object and the blurred object in each picture when the camera lens is set either on the rear object's focal plane or on the front object's focal plane, as shown from each photograph in the optical reconstruction column of Figure 4.

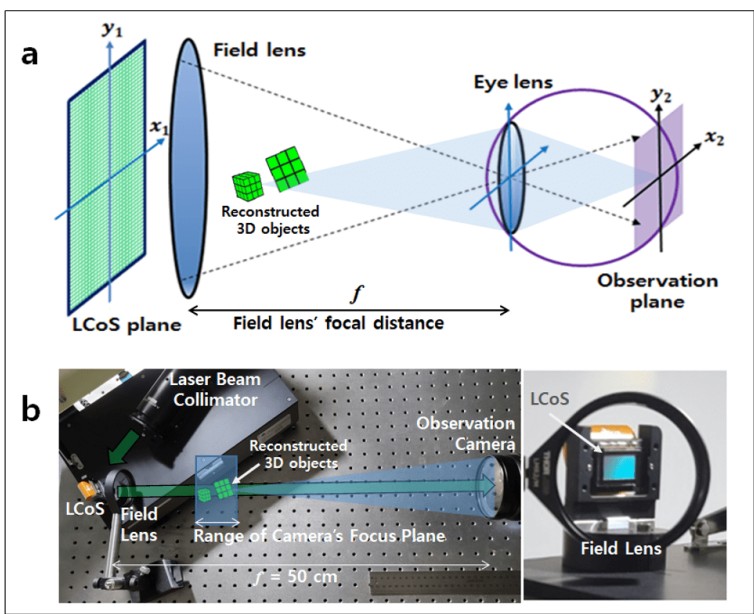

**Figure 5.** Geometry of the optical reconstruction system for holographic 3D observation: (**a**) schematic diagram to be used in numerical simulation, and (**b**) its optical experiment setup. Observer's eye lens in (**a**) is located at the position of the focal length of the field lens, corresponding to the observation position of the Fourier plane away from holographic display (LCoS) in (**b**).

When the holographic 3D images based on the depth map estimated from the proposed deep learning model are observed, there is a weak blurring in comparison with images based on the ground truth depth map, as shown in Figure 4. This is because the difference between depth values of the two objects on the basis of estimation has a minute deviation from that between the depth values of the two objects on the basis of a reference (ground truth). However, the photographs to monitor reconstructed scenes in Figure 4 indicate that this actual deviation is small enough to well support the accommodation effect on holographic 3D scenes for both deep learning and ground truth cases; when an object between two objects exists on the camera's focus, its photographic image is clearly sharp, while an object that is out of focus is completely blurred. Results to observe reconstructed 3D scenes in the other objects are presented in Figure S2.

4.2.2. Comparison of the Proposed Model (HDD) with Previous Models (Using Four Kinds of Objects Used in the Course of the Training Phase)

To qualitatively evaluate the depth map estimation performance of the HDD, we not only compared the depth map estimations to verify the improvement of the proposed HDD model over the CDD model (reference model), but we also compared the CGH reconstruction results of these two models. Furthermore, to increase the reliability of the proposed HDD model, we also compared the estimated depth map/image quality of the holographic 3D reconstruction of our proposed model with another model called AdaBins (state-of-the-art model), which was recently developed by Bhat et al. [31]. In this experimental verification, we included the results of the *ACC* (accuracy test) of CGH images. The AdaBins model, as a competitive model, was also trained using the same training data set that we used for both HDD and CDD models. We mainly focused on

comparing depth map estimation results and CGH's reconstruction results from HDD with those from AdaBins.

Figure 6 shows the depth map estimation results from HDD, AdaBins, and CDD, along with the ground truth depth map for the torus. Figure 7 shows the 3D images reconstructed from CGH for ground truth, HDD, AdaBins, and CDD. When we move the observer's focal point from one object to another for each given scene, as shown in Figure 7, a typical focus–defocus effect is clearly observed between the two objects at the same viewpoint. This effect manifests the accommodation by observing the holographic 3D images reconstructed from CGHs.

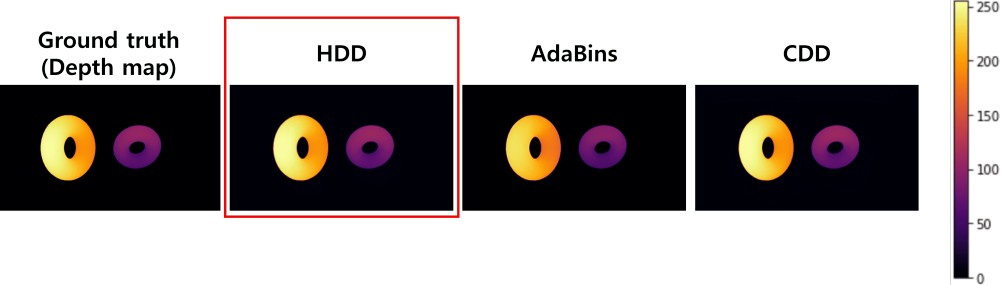

**Figure 6.** Comparison of depth map estimations for the torus. The red box indicates the result from the proposed HDD model. Information close to the camera are represented as brighter, and information far to the camera are represented as darker. Results of depth map estimations with other objects can be seen in Figure S3. The data set for the ground truth depth map is created using Autodesk Maya 2018, Maya Software (https://www.autodesk.com/products/maya/overview, accessed on 17 September 2022).

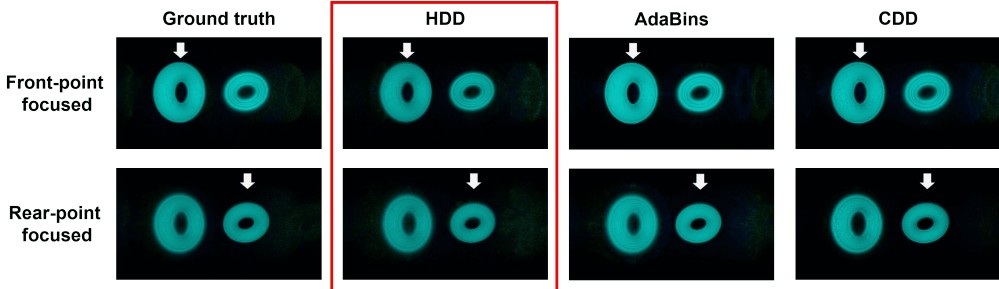

**Figure 7.** Comparison of numerical 3D reconstruction images from CGHs based on estimated depth maps for the torus. (Each focused object is marked by an arrow. The red box indicates the result from the proposed HDD model. Numerical 3D reconstruction results with other objects can be seen in Figure S4).

To quantitatively compare the quality of CGH from the estimated depth map with that of CGH from the ground truth depth map, we carried out a performance evaluation using the *ACC*, which is defined as

$$ACC = \frac{\sum_{r,g,b}(I \cdot I')}{\sqrt{\left[\sum_{r,g,b} I^2\right]\left[\sum_{r,g,b} I'^2\right]}} \tag{3}$$

where $I$ is the brightness of each color in the CGH image obtained using RGB image and the depth map, estimated by each deep learning model, and $I'$ is the brightness of each color in the CGH image obtained using RGB image and the ground truth depth map. When the estimation result and ground truth are identical, or $I = kI'$ ($k$ is a positive scalar), $ACC = 1$. When a mismatch occurs between them, $0 \leq ACC \leq 1$.

We synthesized and reconstructed CGHs using the depth maps estimated from HDD/AdaBins/CDD and the ground truth. For these three types of models, Figure 8 shows

the *ACC* trend of CGHs to the observer's rotation angles for the torus, where the individual CGH is generated using the depth map result estimated from each deep learning model.

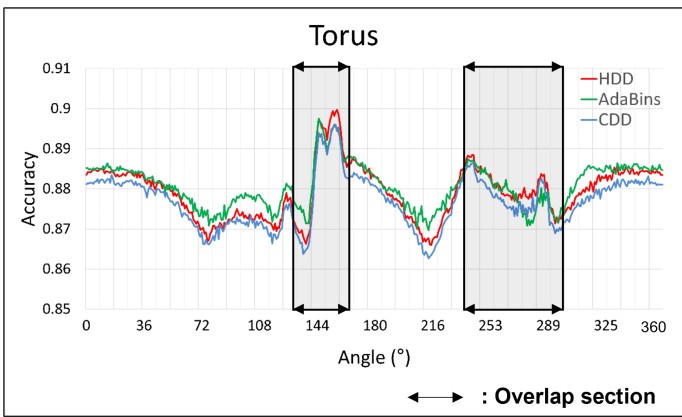

**Figure 8.** ACC trend of CGHs generated using the estimated depth map of each model to the observer's rotation angles for the torus. The *x*-axis represents the observation angle and the *y*-axis represents the *ACC* of CGH. Gray-colored boxes indicate regions where two objects overlap for each 360-degree video scene. (Results of the other objects can be seen in Figure S5).

There are some sections where AdaBins' CGH *ACC* is slightly higher than that of HDD. These sections are mainly located in zones where the two objects do not overlap. However, there are other sections where the CGH *ACC* of HDD is higher than that of AdaBins. These sections are mainly located in the regions where the two objects overlap (gray-colored boxes in Figure 8).

We first checked the depth map estimation results of the HDD and AdaBins models. We then monitored the holographic 3D reconstruction results to determine the different accuracies of the CGHs generated from these two models in two different scenarios. Figures 9 and 10 show the depth map estimation results and the CGH reconstruction results for a typical section where two objects overlap, respectively. When observing the overlapped sections in each case for the torus, cube, and cone, as shown in Figures 9 and S6, we find that the boundary between the front and rear objects is neat in the depth estimation result of HDD, whereas it is uneven in the depth estimation result of AdaBins. In addition, the depth estimation result of AdaBins shows that the area of the front object (brighter area) invades the area of the rear object (darker area). This indicates that the HDD model performs both near- and far-distance estimation tasks well without a shape distortion problem, whereas the AdaBins model estimates the object located at a farther distance to be located at a closer distance.

Furthermore, we observed that the AdaBins' inaccuracy of the depth map estimation near the boundary region overlapping between the front and back objects is again confirmed through the image quality test of a holographic 3D reconstruction. Figure 10 shows the holographic 3D reconstruction results simulated within the typical section in which the two objects overlap. In the case of either a front-point or rear-point focused reconstruction, the overlapped profile in Figure 10 shows that the shape-spreading/distortion phenomenon is more pronounced in 3D reconstruction results based on the estimated depth map of AdaBins in comparison with those based on the estimated depth map of HDD.

In 360-degree holographic video content, which represents dynamic movements among various objects, the capability to express the exact boundary between two objects is critical to holographic 3D image quality. In this aspect, although AdaBins is finely superior to HDD when the objects do not overlap, it is not suitable in a scenario where they overlap for 360-degree digital holographic movies because the AdaBins model does not accurately perform the depth map estimation task. Conversely, it can be seen that the HDD model estimates that depth map information—to a reasonable extent—provides general video content in scenarios even where two objects do not overlap. Overall, we found that HDD is

superior to AdaBins in terms of conserving the very detailed original shape information within the region where two objects overlap, as demonstrated in Figures 9 and 10.

From the experiment, we concluded that the deep learning-based HDD model proposed in the study can reasonably provide an adequate quality of depth map estimation, which can be used to generate CGH for 360-degree holographic video content, especially without noticeable shape distortion of the holographically reconstructed 3D image.

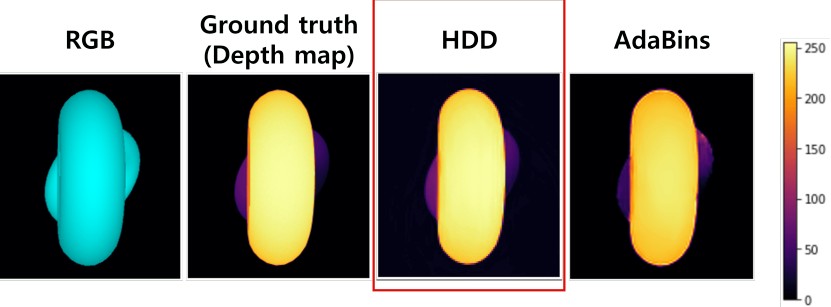

**Figure 9.** Depth map estimation comparison to the ground truth depth map for a typical section where two objects overlap. The red box indicates the result from the proposed model. Results of the other objects can be seen in Figure S6. Data set for RGB color image and ground truth depth map is created using Autodesk Maya 2018, Maya Software (https://www.autodesk.com/products/maya/overview, accessed on 17 September 2022).

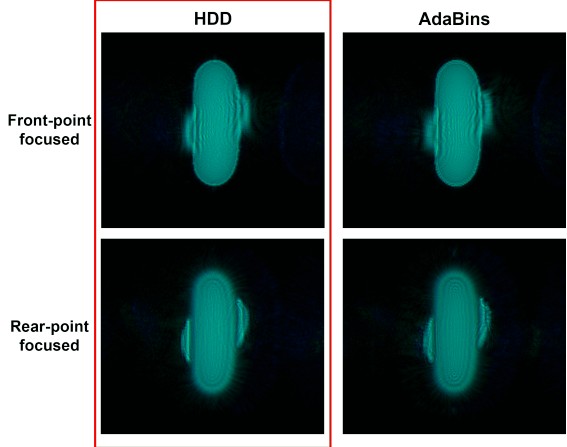

**Figure 10.** CGH's reconstruction results for a typical section where two objects overlap. (The red box indicates the result from the proposed model. Results of the other objects can be seen in Figure S7). In the case of AdaBins, it is observed that the shape of the holographically reconstructed object is distorted because of the severe inaccuracy of depth map estimation by the AdaBins model near the boundary regions between objects.

4.2.3. Comparison of the Proposed Model (HDD) with Previous Models (Using Two Kinds of Complicated Objects Not Used in the Course of the Training Phase)

Unlike the previous experiment performed in Section 4.2.2, we present another experiment of depth map estimation performed using new, complicated objects that are not used in the course of the training phase so that we can demonstrate the superiority of the proposed model (HDD) over AdaBins and CDD. In addition, we synthesized CGH with the estimated depth map and then reconstructed a holographic 3D image to compare the depth map estimation performance of the proposed model with the performance of previous models. For this experiment, we chose an icosahedron and a dodecahedron as the solid figures, which are more complicated than the four kinds of objects used for the training phase. RGB color images of these two objects not used in the training process were used as input data for each model and then the corresponding depth map images were

estimated from each model. The depth maps, as shown in Figure 11, were estimated under the same conditions as the weights of each model which had been obtained in the course of the training phase based on the four pre-existing objects (torus, cube, cone, and sphere).

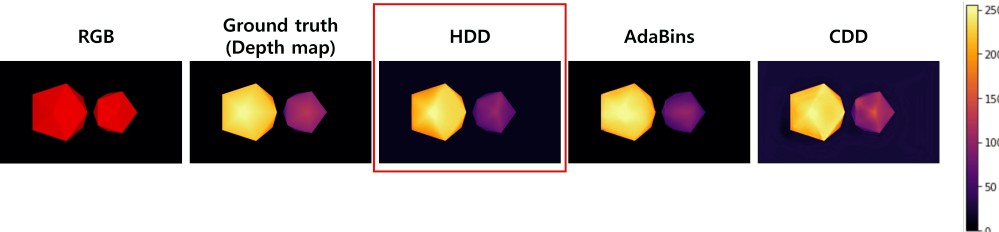

**Figure 11.** Depth map estimation results for icosahedron, which was not used in the course of the training phase. The red box indicates the result from the proposed model. Results of the dodecahedron can be seen in Figure S8. The data set for RGB color image and ground truth depth map for a pair of these solid objects was created using Autodesk Maya 2018, Maya Software (https://www.autodesk.com/products/maya/overview, accessed on 17 September 2022).

In Figure 11, it can be visually confirmed that the edge expressions in individual objects from HDD or CDD are more accurate than the results from AdaBins for unlearned, complicated objects. To compare the results of HDD with those of CDD, we calculated values of the *PSNR* for these two objects (dodecahedron and icosahedron): the *PSNR* of HDD is 73.02 dB for icosahedron and 72.96 dB for dodecahedron, while the *PSNR* of CDD is 70.25 dB for icosahedron and 71.23 dB for dodecahedron.

Figure 12 shows the 3D images reconstructed from CGH for ground truth, HDD, AdaBins, and CDD. As shown in Figure 12, a typical focus–defocus effect can be observed between the two objects at the same viewpoint; the accommodation effect is typically observed when holographic 3D images are reconstructed from CGHs.

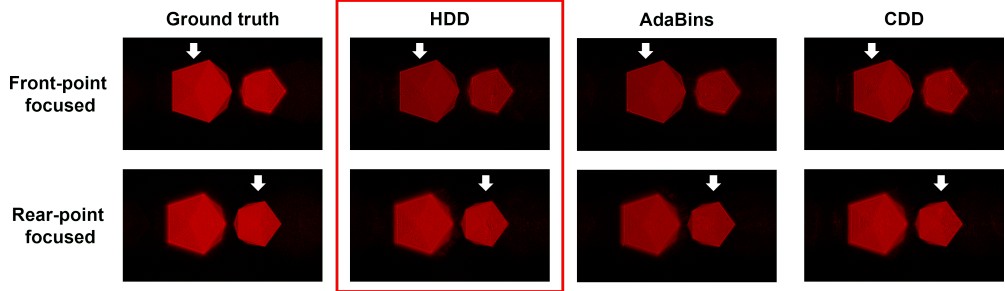

**Figure 12.** Holographic 3D reconstruction from CGH using each set of RGB color and depth map given in Figure 11. The focused object between two objects to confirm the accommodation effect is marked by an arrow. (The red box indicates the result from the proposed model. Results of the dodecahedron can be seen in Figure S9).

Figure 13 shows the *ACC* trend of CGHs to the observer's rotation angle for the unlearned icosahedron. The individual CGH is generated using the depth map result estimated from each deep learning model. For the section where two objects do not overlap, the *ACC* of each model shows a similar trend. However, some local spots exist (ranging from angles 72° to 90°, and 216° to 234° in Figure 13) in which the *ACC* of Adabins is significantly lower than that of HDD. Moreover, for the section where two objects overlap (the gray boxes in Figure 13), the *ACC* of HDD is higher than that of AdaBins. From the *ACC* test of CGHs in the case of unlearned, complicated objects, we found that the *ACC* trend of CGHs that originate from AdaBins shows inconsistent behavior during a cycle of the 360-degree scenes, and the AdaBins model is inferior to the HDD model in the *ACC* of CGHs within the regions in which two objects overlap.

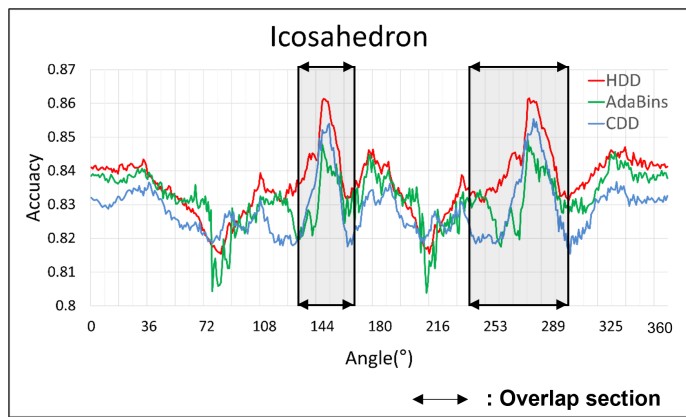

**Figure 13.** ACC trend of CGHs generated using the depth map estimated from each model to the observation's rotation angle for the shape of an icosahedron. The *x*-axis represents the observation's rotation angle and the *y*-axis represents the *ACC* of CGH synthesized from each model. Gray-colored boxes indicate regions where two objects overlap for each 360-degree video scene. (Results of the dodecahedron can be seen in Figure S10).

Figure 14 shows the depth map estimation results from HDD and from AdaBins for a typical case of the section where two objects overlap. From the shapes of the icosahedron, as presented in Figure 14, we observe that near the boundary region between the front and rear objects, the depth map estimation quality of HDD is much better than that of AdaBins. This result indicates that the proposed HDD model carries out both near- and far-distance estimation tasks from the camera well without a distorted shape. Figure 15 shows the holographic 3D reconstruction results simulated from CGHs, which were generated using the same depth map estimations as we obtained in Figure 14. In the case of either a front-point or rear-point focused reconstruction, it is observed that the shape distortion in the case of AdaBins is more conspicuous than that of an HDD. The reason is that the inaccuracy of depth map estimation by the AdaBins model becomes more severe near the boundary region between objects as the object's shape becomes more complicated. In conclusion, through the experiments of depth map estimation, CGH synthesis, and reconstruction with complicated objects not used in the training phase, we found out that the deep learning-based proposed model (HDD) is superior to AdaBins and CDD in the ability to estimate depth map for each 360-degree scene regarding unlearned, more complicated objects.

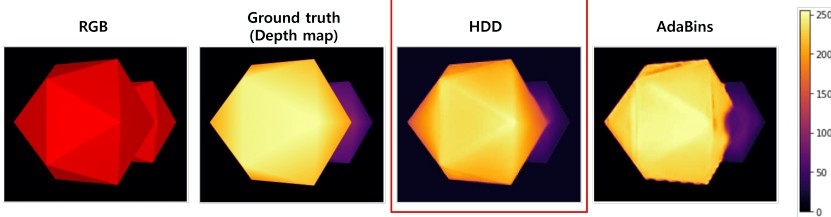

**Figure 14.** Depth map estimation comparison with respect to the ground truth depth map in a typical section where two objects overlap. Each of these solid objects was an icosahedron, of which was not used in the course of the training phase. The red box indicates the result from the proposed model. Results of the dodecahedron can be seen in Figure S11. The data set for RGB color image and ground truth depth map was created using Autodesk Maya 2018, Maya Software (https://www.autodesk.com/products/maya/overview, accessed on 17 September 2022).

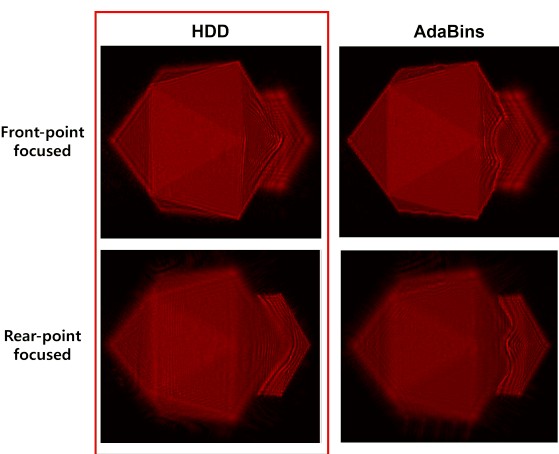

**Figure 15.** Reconstruction results from CGHs in a typical section where two objects overlap, as seen in Figure 14. (The red box indicates the result from the proposed model. Results of the dodecahedron can be seen in Figure S12).

### 4.2.4. Further Comparison of the Proposed Model (HDD) with Previous Models (Using Complicated Object)

We present a further experiment of depth map estimation performed using more complicated objects that are not used in the course of the training phase so that we can demonstrate the superiority of the proposed model (HDD) over AdaBins and CDD in applications for 360-degree digital holographic content. We synthesized CGH with the estimated depth map and then reconstructed a holographic 3D image to compare the depth map estimation performance of the proposed model with the performance of previous models. For this experiment, we chose an amorphous rock, which is as complicated as the level of real objects. Since Section 4.2.4 is part of additional experiments, the data used in this section were not included in the total number of data—9832 images—mentioned in this paper. The depth maps, as shown in Figure 16, were estimated under the same conditions as the weights of each model which had been obtained in the course of the training phase based on the four pre-existing objects (torus, cube, cone, and sphere).

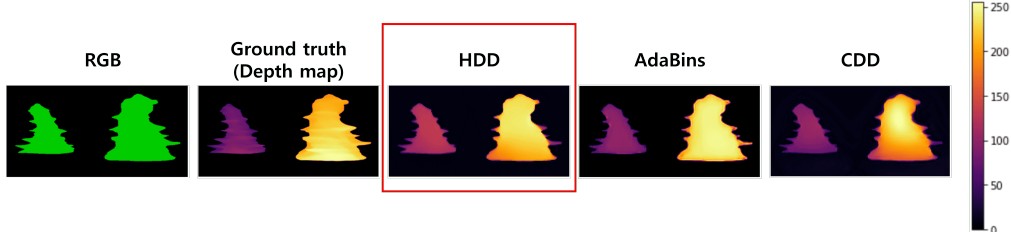

**Figure 16.** Depth map estimation results for a new scene which was composed of a pair of amorphous solid rocks. The red box indicates the result from the proposed model (HDD) in comparison with other models. The 3D model of each amorphous rock was provided from TurboSquid, https://www.turbosquid.com/ko/3d-models/cave-rock-02-base-model-1944210, accessed on 17 September 2022). Data set for RGB color image and ground truth depth map for a pair of these solid objects was created using Autodesk Maya 2018, Maya Software (https://www.autodesk.com/products/maya/overview, accessed on 17 September 2022).

In Figure 16, although the approximate trends of HDD and CDD are similar, it can be visually confirmed that HDD is closer to the ground truth depth map compared to CDD. It can be visually confirmed that the AdaBins estimated objects to be brighter than the ground truth depth map. To compare the results of HDD with those of CDD, we calculated values of the *PSNR* for an amorphous rock: the *PSNR* of HDD was 27.65 dB, while the *PSNR* of CDD was 26.27 dB.

The observational camera's focus was placed on the head of the target object between the two objects reconstructed from CGH, as marked with an arrow in Figure 17. Figure 17 shows the 3D images reconstructed from CGH for ground truth, HDD, AdaBins, and CDD. As shown in Figure 17, a typical focus–defocus effect can be observed between the two objects at the same viewpoint; the accommodation effect is typically observed when holographic 3D images are reconstructed from CGHs.

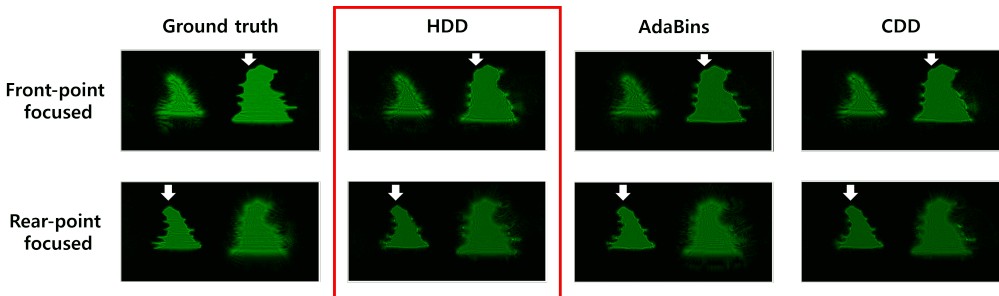

**Figure 17.** Holographic 3D reconstruction from CGH using each set of RGB color and depth map given in Figure 16. The focused object between two objects to confirm the accommodation effect is marked by an arrow. (The red box indicates the result from the proposed model).

Figure 18 shows the *ACC* trend of CGHs to the observer's rotation angle for unlearned amorphous rock. The individual CGH is generated using the depth map result estimated from each deep learning model. For the section where two objects do not overlap, the *ACC* of each model shows a similar trend. However, for the section where two objects overlap (the gray boxes in Figure 18), the *ACC* of HDD is significantly higher than that of AdaBins. Similar to the simulation results in Section 4.2.3, from the *ACC* test of CGHs in the case of unlearned, complicated objects, we found that the *ACC* trend of CGHs that originate from AdaBins shows inconsistent behavior during a cycle of the 360-degree scenes, and the AdaBins model is inferior to the HDD model in the *ACC* of CGHs within the regions in which two objects overlap.

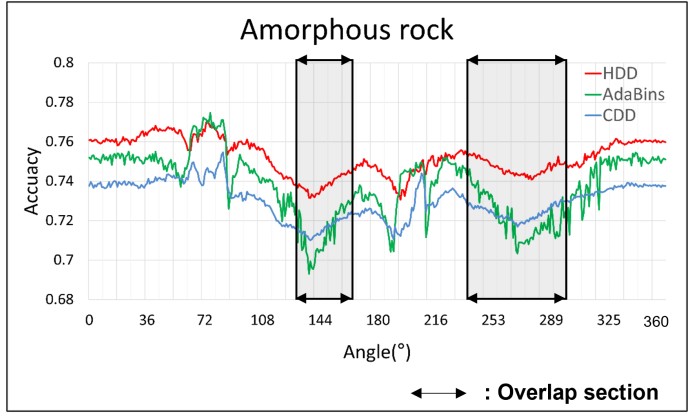

**Figure 18.** ACC trend of CGHs generated using the depth map estimated from each model to the observation's rotation angle for the shape of an amorphous rock. The *x*-axis represents the observation's rotation angle and the *y*-axis represents the *ACC* of CGH synthesized from each model. Gray-colored boxes indicate regions where two objects overlap for each 360-degree video scene.

Figure 19 shows the depth map estimation results from HDD and from AdaBins for a typical case of the section where two objects overlap. From the shapes of the amorphous rock, as presented in Figure 19, we observe that near the boundary region between the front and rear objects, the depth map estimation quality of HDD is much better than that of AdaBins. This result indicates that the proposed HDD model carries out both near- and far-distance estimation tasks from the camera well without a distorted shape.

Figure 20 shows the holographic 3D reconstruction results simulated from CGHs, which were generated using the same depth map estimations as we obtained in Figure 19. In the case of either a front-point or rear-point focused reconstruction, it is observed that the shape distortion in the case of AdaBins is more conspicuous than that of an HDD. The reason is that the inaccuracy of depth map estimation by the AdaBins model becomes more severe near the boundary region between objects as objects' shape becomes more complicated. In conclusion, through the experiments of depth map estimation, CGH synthesis and reconstruction with complicated objects not used in the training phase, similar to the simulation results in Section 4.2.3, we found out that the deep learning-based proposed model (HDD) is superior to AdaBins and CDD in the ability to estimate depth map for each 360-degree scene regarding unlearned, more complicated objects.

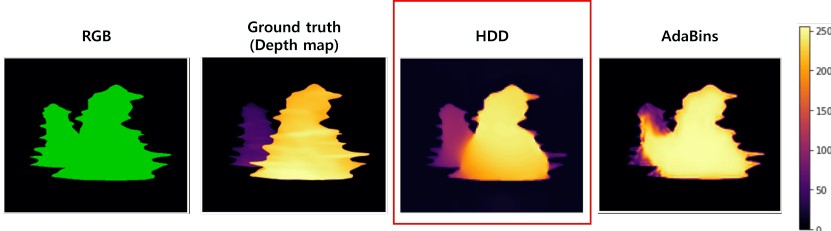

**Figure 19.** Depth map estimation comparison with respect to ground truth depth map in a typical section where two objects overlap. Each of these solid objects was a new amorphous 3D rock which was not used in the course of the training phase. The red box indicates the result from the proposed model. The 3D model of each amorphous rock was provided from TurboSquid, https://www.turbosquid.com/ko/3d-models/cave-rock-02-base-model-1944210, accessed on 17 September 2022. the data set for RGB color image and ground truth depth map was created using Autodesk Maya 2018, Maya Software (https://www.autodesk.com/products/maya/overview, accessed on 17 September 2022).

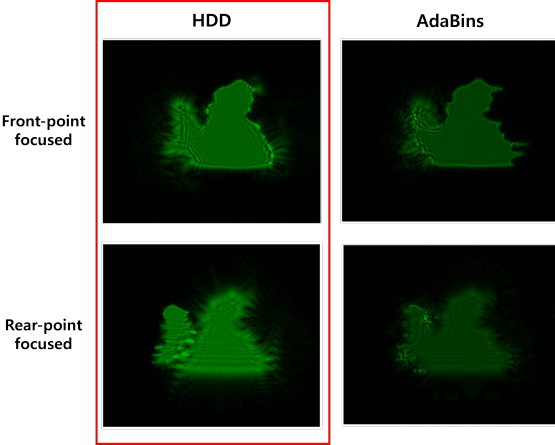

**Figure 20.** Reconstruction results from CGHs in a typical section where two objects overlap, as seen in Figure 19. The observational camera's focus was placed on the head of the target object between the two objects reconstructed from CGH. (The red box indicates the result from the proposed model).

## 5. Discussion and Conclusions

In this paper, we proposed and demonstrated a novel CNN model that learns depth map estimation from missing viewpoints, especially well-fit for holographic 3D. The proposed model, which we call HDD, uses only *MSE* for better depth map estimation performance in comparison with the CDD, which uses *SSIM* 90% and *MSE* 10% as the loss function.

We designed and prepared 9832 multi-view images with a resolution of 640 × 360. In the proposed model, HDD estimated depth maps by extracting features and up-sampling.

The weights were optimized using the *MSE* loss function. For quantitative assessment, we compared the estimated depth maps from HDD with those from CDD using the *PSNR*, *ACC*, *RMSE*. As shown in Figure 3, the proposed HDD model is numerically superior to CDD in terms of *PSNR*, *ACC*, and *RMSE*. In addition, as shown in Table 4, although HDD is numerically inferior to CDD on the metrics of absolute relative error (*Abs rel*), it is superior to CDD on the metrics of squared relative error (*Sq rel*) and *LRMSE*. To qualitatively evaluate the depth map estimation performance of the HDD, we not only compared the depth map estimation results to verify the improvement of the proposed HDD model over the CDD model, but also compared the CGH reconstruction results of these two models. Furthermore, to increase the reliability of the proposed model, we also compared the estimated depth map image quality of the holographic 3D reconstruction of our proposed model with another competitive model, AdaBins. Through the experiments of depth map estimation, CGH synthesis, and reconstruction with both objects used in the training and complicated objects not used in the training, we found out that HDD is more suitable than AdaBins in scenarios such as objects overlapping for 360-degree digital holographic movies. Furthermore, we found that HDD is superior to AdaBins and CDD in the ability to estimate depth map for each 360-degree scene regarding unlearned, more complicated objects.

The contributions of this study are as follows: First, we demonstrate the ability of our proposed HDD to learn and produce depth map estimation with high accuracy from multi-view RGB images. Second, we prove the feasibility of applying deep learning-based, estimated depth maps to synthesize CGHs, with which we can quantitatively evaluate the degree of accuracy in the performance of our proposed model for holography. Third, we illustrate the effectiveness of CGHs synthesized via the proposed HDD through direct numerical/optical observations of holographic 3D images.

The limitations of the proposed model covered in this study are the minute residual images near the border area of each object and the weak background noise in the estimated depth maps. To overcome these issues, one needs to find approaches to place relatively large weights on these border areas and then enhance precision estimations of these areas. It is also worth mentioning that the image resolution and extraction speed of deep learning-based depth maps can be improved by adjusting the model's parameters, such as filters and filter sizes, and then optimizing the ratio of training/testing data. Moreover, we note that only the diffraction efficiency element, that is, the direct observation of the reconstructed holographic 3D image, was used as a comparative measure for the quality of the H3D image in this study. We plan to supplement this measure through further analysis, considering parameters such as the contrast ratio of intensity, clearness, and distortion to evaluate H3D images. We plan a future study to optimize the interval of missing viewpoints. Currently, without the standard appropriate for missing viewpoints, situations in various viewpoints have been used to generate 360-degree holographic video content. Future studies to optimize the interval of missing viewpoints are expected to contribute to AR, VR, and holographic metaverse applications, leading to a low computational-cost method in providing realistic 360-degree holographic content.

**Supplementary Materials:** The following supporting information can be downloaded at: https://www.mdpi.com/article/10.3390/app12199432/s1, Figure S1: PSNR trend with respect to loss function (MSE-SSIM) coefficient; Figure S2: Results of CGH reconstruction for cube, cone, and sphere; Figure S3: Depth map estimation result comparison for cube, cone, and sphere; Figure S4: Numerical CGH reconstruction result comparison for cube, cone, and sphere; Figure S5: ACC trend of CGHs generated using the estimated depth map of each model to the observer's rotation angles for objects; Figure S6: Depth map estimation comparison to ground truth depth map for a typical section where two objects overlap; Figure S7: CGH's reconstruction results for a typical section where two objects overlap; Figure S8: Depth map estimation results for dodecahedron, which is not used in the course of the training phase; Figure S9: Holographic 3D reconstruction from CGH for objects not used in the course of training phase to confirm the accommodation effect; Figure S10: ACC trend of CGHs generated using the depth map estimated from each model to the observation's rotation angle for

the shape of a dodecahedron; Figure S11: Depth map estimation comparison with respect to ground truth depth map in a typical section where two objects overlap with the solid shape of dodecahedron which is not used in the course of the training phase; Figure S12: CGH's reconstruction results in a typical section where two objects overlap with the solid shapes of dodecahedron, which is not used in the course of the training phase; Video S1: Results of numerical CGH reconstruction for entire viewpoints; Video S2: Result of optical CGH reconstruction for entire viewpoints.

**Author Contributions:** Conceptualization, H.K., M.Y. and C.K.; methodology, H.K., M.Y. and C.K.; software, H.K., H.L. and M.J.; validation, H.K., M.Y. and C.K.; formal analysis, H.K., H.L. and M.J.; investigation, H.K., H.L. and M.J.; resources, H.K. and Y.L.; data curation, H.K. and Y.L.; writing—original draft preparation, H.K., H.L. and M.J.; writing—review and editing, H.K., M.J. and M.Y.; visualization, H.K.; supervision, M.Y. and C.K.; project administration, M.Y. and C.K.; funding acquisition, M.Y. and C.K. All authors have read and agreed to the published version of the manuscript.

**Funding:** This research was funded by the Ministry of Science and ICT (MSIT, South Korea) grant number V0408-20-1001. The APC was funded by authors.

**Institutional Review Board Statement:** Not applicable.

**Informed Consent Statement:** Not applicable.

**Data Availability Statement:** Not applicable.

**Conflicts of Interest:** The authors declare no conflicts of interest.

## Abbreviations

The following abbreviations are used in this manuscript:

| | |
|---|---|
| CGH | Computer-generated holograms |
| CNN | Convolutional neural network |
| CDD | Conventional dense depth |
| AR/VR | Augmented reality/Virtual Reality |
| HDD | Holographic dense depth |
| MSE | Mean squared error |
| SSIM | Structural Similarity |
| PSNR | Peak signal-to-noise |
| ACC | Accuracy |
| SLM | Spatial light modulator |
| LCoS | Liquid crystal on silicon |

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
