# Peer review of "High-Precision Depth Map Estimation from Missing Viewpoints for 360-Degree Digital Holography"

_applsci, doi:10.3390/app12199432_

Round 1

Reviewer 1 Report

The authors propose and validate a novel model for recovery of depth maps of 3D objects from missing view points for generating holographic 3D images and video. The manuscript is well written and presented including the figures and supplemental material. The topic overall is quite interesting and may be important in certain applications. The algorithm is described and validated, although quite simple objects were used in validation. In my opinion the novelty of the manuscript and its originality allow to publish it in Applied Sciences, although several questions related with validation, data presentation and relevance of the problem should be addressed before the publication.

·    1) I miss the discussion related with the number of missing view points from which the depth information is reconstructed. Also I believe that location of such viewpoints can be also very important. Figure 1(a) suggests that the missing viewpoints are located in quite a large sector of about 90 degrees. Is it real situation in which the data of the object can be still reconstructed from?

·     2) Almost all of the studies shapes (torus, cube, cone, sphere, dodecahedron, and icosahedron) are convex objects and it seems to me that all of them do not have small details. Moreover, all of them somehow has a certain degree of symmetry. On the other hand real objects are typically more complex, they may have a certain small details and which is more important, they can be not convex objects (those which curved inward rather than outward). I wonder if the authors could perform reconstruction of such a more complex object and validate their technique.

·  3) When presenting depth maps of the objects (both ground truth and recovered depth maps), probably it would be useful to somehow color-code the actual depth, so that the readers could easily estimate quality of the recovered image. For example the ground-truth dodecahedron in figure 14 seems to be homogeneous white color, so it seems that its depth doesn’t vary at all. I believe that if the grayscale depth mapping would be replaced with colormap depth mapping the results obtained would be more evident.

·       4) Although the authors have described experimental setup in subsection 4.2.1 I believe that it would be useful to indicate focal distances in fig. 5(b) and to indicate the focus plane of the camera.

·   5) The authors propose a method to estimate depth maps of missing view points of 3D objects. As far as I understand they indicate two major applications of their method: generation of CGH (Computer-generated holograms) and Augmented reality/Virtual Reality. Moreover they demonstrate the technique on simulated 3D objects such as torus, cube, cone etc. However I don’t understand why do the missing points of simulated objects may actually exist? As far as I understand, once the object is known and 3D-simulated one can easily calculate its appearance and depth map from any view points. It’s a bit hard to understand to me how a-priori simulated objects may have a missing view points? Could you please add some discussion of this topic.

Reviewer 2 Report

In this paper, the authors proposed and researched the method of depth map estimation from missing viewpoints for 3D CGH. To demonstrate the advantages of the method, the authors compare their HDD method with the well-known CDD method (Fig.3). The method allows to reduce the error (MSE) by 2 or more times. This work is suitable for publication in the journal Appl Sci, as it is not aimed at obtaining new knowledge, but is aimed at improving known methods through deep learning. It is difficult to assess the scientific significance of this work, since there are only two simple equations in the work, (1) and (2). Moreover, the functions H and Li are not defined in equation (1). Nevertheless, the method proposed in the work of HDD has been studied in detail and its advantage over the known method has been shown. Therefore, I recommend publishing this work.
